

# The Lifshitz nature of the transition between the gap and gapless states of a superconductor

Yuriy Yerin[1], Caterina Petrillo[1] and A. A. Varlamov[2]

**1** Dipartimento di Fisica e Geologia, Universitá degli Studi di Perugia,
Via Pascoli, 06123 Perugia, Italy
**2** CNR-SPIN, via del Fosso del Cavaliere, 100, 00133 Roma, Italy

## Abstract

It is demonstrated that the known for a long time transition between the gap and the gapless states in the Abrikosov-Gor'kov theory of a superconductor with paramagnetic impurities is of the Lifshitz type, i.e. of the $2\frac{1}{2}$ order phase transition. We reveal the emergence of a cuspidal edge at the density of states surface $N(\omega, \Delta_0)$ ($\Delta_0$ is the value of the superconducting order parameter in the absence of magnetic impurities) and the occurrence of the catastrophe phenomenon at the transition point. We study the stability of such a transition with respect to the spatial fluctuations of the magnetic impurities critical concentration $n_s$ and show that the requirement for validity of its mean field description is unobtrusive: $\nabla(\ln n_s) \ll \xi^{-1}$ (here $\xi$ is the superconducting coherence length). Finally, we show that, similarly to the Lifshitz transition, the transition between gap and gapless states should be accompanied by the corresponding singularities. For instance, the superconducting thermoelectric effect has a giant peak exceeding the normal value of the Seebeck coefficient by the ratio of the Fermi energy and the superconducting gap. The concept of the experiment for the confirmation of $2\frac{1}{2}$ order transition nature is proposed. The obtained theoretical results can be applied for the explanation of recent experiments with lightwave-driven gapless superconductivity, for the new interpretation of the disorder induced transition $s_{\pm}$-$s_{++}$ states via gapless state in multi-band superconductors, for better understanding of the gapless color superconductivity in quantum chromodynamics, the string theory.

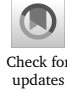

# 1  Introduction

In 1960, two seminal papers were published almost simultaneously, which gave rise to new directions in the research fields of superconductivity and fermiology [1,2].

In the first paper, Abrikosov and Gor'kov (AG), extending the Bardeen-Cooper-Schrieffer (BCS) theory to the case of a superconducting alloy containing paramagnetic impurities, demonstrated that the original BCS identification of the phenomenon of superconductivity with the presence of the gap in the quasiparticle spectrum is too restrictive, and, under some conditions, gapless superconductivity can exist. According to the AG theory [1,3], the transition between gap and gapless regimes was governed by the concentration of paramagnetic impurities and the properties of such superconducting system were studied in the mean-field approximation. Gapless superconductivity occurs in the very narrow interval of paramagnetic impurity concentrations $0.912\, n_c < n < n_c$, where $n_c$ is the concentration that completely suppresses the supercurrent flow. Later, it was recognized that the gapless regime in a superconductor can be induced by different mechanisms breaking the time-reversal symmetry: magnetic field [4], current [4], proximity effect [5] and the light [6]. However, the order of this transition, to the best of our knowledge, was never discussed.

In the second of mentioned above papers I.M. Lifshitz [2] introduced the notion of phase transition of fractional, $2\frac{1}{2}$, order. Also, it was pointed out that by varying some external parameter (pressure or concentration of the isovalent impurities) one can change the number of components of topological connectivity of the Fermi surface (FS), which is accompanied, according to the Ehrenfest terminology [7], by the $2\frac{1}{2}$ order phase transition. Further studies of these, named today as Lifshitz, transitions revealed that they are supplemented by singularities in various properties of the system [8,9].

The ideas proposed 60 years ago remain still requested in modern studies on the stability of current-biased superconducting wires (see [10] and references therein), transformations of the complex heavy fermion Fermi surfaces due to magnetic field effects [11,12], etc. Moreover, the concept of a connection between the topological properties of the different materials exhibiting gapless states and the occurrence of the exotic Lifshitz transitions was recently discussed in literature based on very general topological arguments. Examples are given by Dirac and Weyl materials, and even more exotic systems (see the reviews [13,14]).

In this article we aim at framing these concepts into a unified description and show that the known for a long time transition between gap and gapless superconducting states is the phase transition of the Lifshitz type, i.e. of the $2\frac{1}{2}$ order. This will be proved by a very simple approach, in spirit of the fundamental paper [2], just analyzing the properties of the free energy in a superconductor containing paramagnetic impurities.

Further, we study the requirements on the homogeneity of the paramagnetic impurities

concentration, which is necessary for the validity of the standard mean field approximation in the description of the "gap-gapless" transition used in [1], and prove the stability of the transition with respect to these fluctuations.

Finally, we argue that such a transition would be accompanied by the appearance of singularities in several properties, in particular an anomalous growth of the thermoelectric effect (see [15–18]) close to the critical concentration, which is valuable for the experimental verification of the proposed connection.

It is important to note that we confine our study to the case of a s-wave isotropic superconductor and do not consider unconventional and exotic pairing symmetries.

## 2 Free energy and phase transition

We start from the expression for the free energy close to the transition between gapless and gap regimes at $T = 0$ (see [4, 19]), that is

$$
F_{s-n} = -\frac{N(0)\Delta^2}{2}
\begin{cases}
1 - \dfrac{\pi}{2}\zeta + \dfrac{2}{3}\zeta^2\,, \ \zeta \leqslant 1 \\[2mm]
1 - \zeta \arcsin \zeta^{-1} + \zeta^2\left(1 - \sqrt{1 - \zeta^{-2}}\right) \\[2mm]
\quad -\dfrac{1}{3}\zeta^2\left(1 - \left(1 - \zeta^{-2}\right)^{3/2}\right),\ \zeta > 1\,.
\end{cases}
\tag{1}
$$

$\Delta = \Delta(\tau_s)$ is the order parameter in the presence of impurities ($\Delta \in \mathbb{R}$) and $N(0) = \frac{mp_F}{\pi^2\hbar^3}$ is the density of states (DOS) at the Fermi level. The parameter

$$
\zeta = (\tau_s\Delta)^{-1}\,,
\tag{2}
$$

with $\tau_s$ as the electron spin-flip scattering time due to the presence of paramagnetic impurities, governs the phase transition between the gap and gapless states. Namely, when $\zeta < 1$ the energy gap $\Delta_g$ in the quasiparticle spectrum of a superconductor has a nonzero value, while for $\zeta \geq 1$ the energy gap remains identically equal to zero and the gapless state is realized. At the same time the order parameter $\Delta$ is different from zero and the phenomenon of supercurrent flow occurs. The critical point $\zeta = 1$ separates the gap and the gapless states. We remark that the authors of [1] were the first who pointed out at the importance of making a distinction between the order parameter $\Delta$ and the energy gap $\Delta_g$ existing in the quasiparticles spectrum.

To elucidate what is the order of the phase transition, we first studied the behavior of the free energy (Eq. 1) and its derivatives over the parameter $\zeta$ that drives the transition. It turns out that the free energy together with its first and second derivatives remain continuous function at the transition point $\zeta = 1$. However, the plot of the second derivative $\partial^2 F_{s-n}/\partial\zeta^2$ unambiguously shows the kink at $\zeta = 1$ (Fig. 1). Moreover, from the expression of the third derivative

$$
\frac{\partial^3 F_{s-n}}{\partial\zeta^3} = N(0)\Delta^2
\begin{cases}
0\,, \ \zeta \leqslant 1 \\[2mm]
\dfrac{1}{\zeta^4\sqrt{\zeta^2 - 1}}\,, \ \zeta > 1\,,
\end{cases}
\tag{3}
$$

one can see the occurrence of the characteristic discontinuity in it with the square root singularity from the gapless side. I.e., the situation is completely analogous to the Lifshitz $2\frac{1}{2}$ order phase transitions in metals. The analogy is also confirmed by the DOS dependence on the parameter $\zeta$ driving the transition. It was shown [3, 4, 19] that the quasiparticle DOS of the superconductor $N_s(\omega)$ remains finite at $\omega = 0$ and has a typical cusp for $2\frac{1}{2}$ order phase

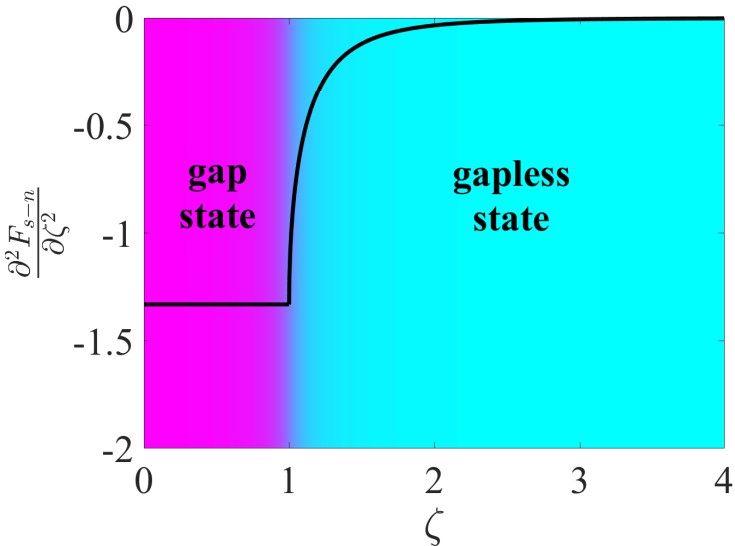

Figure 1: The second derivative of Eq. (1). The kink is clearly observed at $\zeta = 1$. Purple and cyan colors in the background of the plot illustrate separation between gap and gapless states respectively. Weak blur near $\zeta = 1$ represents smearing of the transition due to spacial fluctuations of the magnetic impurities concentration (see the corresponding section of the paper).

transition at $\zeta = 1$

$$N_s(0) = N(0) \frac{\sqrt{\zeta+1}}{\zeta} \sqrt{\zeta - 1}. \tag{4}$$

The appropriate interpretation of the gap-gapless transition can be given studying the transformation of the surface $N(\omega, \Delta_0)$ in the phase space $\omega$-$\Delta_0$, where $\Delta_0$ is the value of the superconducting order parameter in the absence of magnetic impurities. For this purpose we start with the general expression [1,3]

$$N(\omega, \zeta) = N(0)\zeta^{-1} \operatorname{Im} u, \tag{5}$$

where $u$ is given by

$$\frac{\omega}{\Delta} = u\left(1 - \frac{\zeta}{\sqrt{1-u^2}}\right), \tag{6}$$

and the expression which determines the order parameter $\Delta$ at $T = 0$ [1,4]

$$\ln\left(\frac{\Delta}{\Delta_0}\right) = \begin{cases} -\dfrac{\pi}{4}\zeta, & \zeta \leqslant 1 \\ -\operatorname{arccosh}\zeta - \dfrac{1}{2}\left(\zeta \arcsin \zeta^{-1} - \sqrt{1-\zeta^{-2}}\right), & \zeta > 1. \end{cases} \tag{7}$$

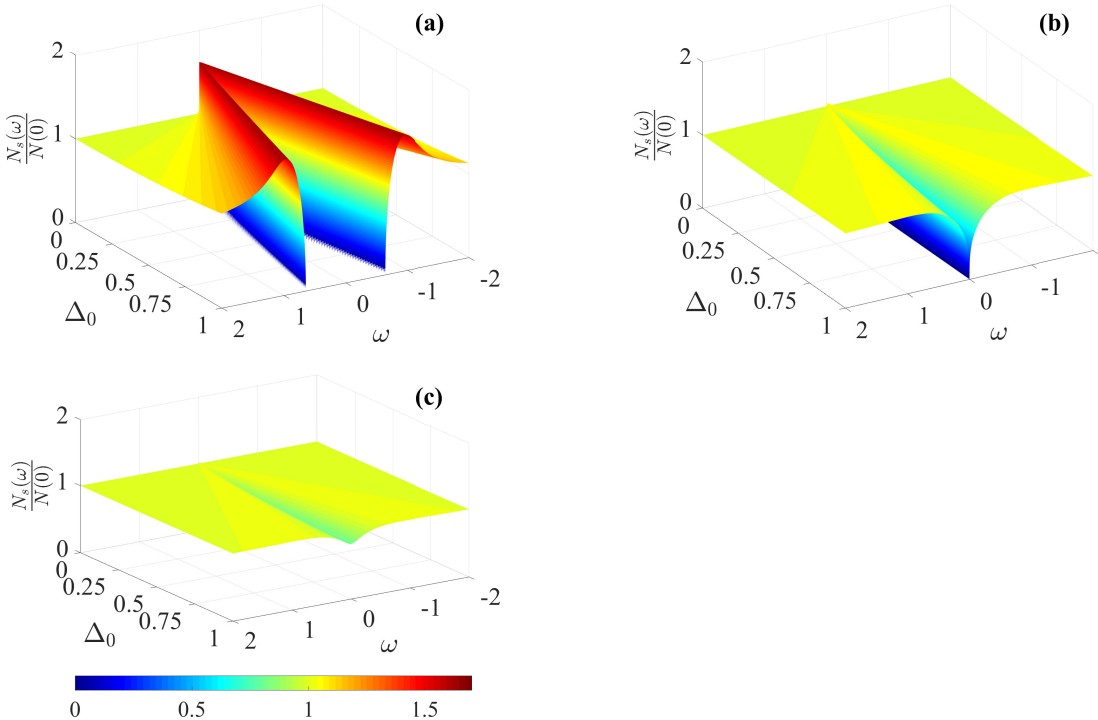

Figure 2: Evolution of the DOS in the $\omega$-$\Delta_0$ space from the gap state with $\zeta = 0.1$ (a), through the collapse of the energy gap $\zeta = 1$ (b), to the gapless state with $\zeta = 1.75$ (c). Arbitrary units for $\omega$ and $\Delta_0$ are used in the legends.

Based on Eqs. (5) and (7) one can track the evolution of the function $N(\omega, \Delta_0)$ in the $\omega$-$\Delta_0$ phase space via three stages that are characterized by dissimilar surfaces for $\zeta < 1$, $\zeta = 1$ and $\zeta > 1$ (see Fig. 2). The first stage corresponds to the gap state with $\zeta < 1$ and with the characteristic narrowing hollow between two glued sheets of the DOS surfaces at $\Delta_g = 0$ (Fig. 2a). The collapse of the energy gap when $\zeta = 1$, and the subsequent emergence of a topological feature in the form of the pleat known as a cuspidal edge at $\omega = 0$, is shown in Fig. 2b. This feature allows to speculate about the occurrence of the catastrophe phenomenon in the $\omega$-$\Delta_0$ space over the gap-gapless phase transition [22–25]. Finally, the last stage with $\zeta > 1$ corresponds to the gapless state with the gradual degradation of the DOS curved surface to a plane for $\zeta \to \infty$ (Fig. 2c).

In this representation one can freely "travel" over the each surface $N(\omega, \Delta_0)$ by changing the variables $\omega$ and $\Delta_0$ while keeping $\zeta = \zeta(\Delta_0, \tau_s) = const$ and adjusting the value of $\tau_s$ for each $\Delta_0$ to satisfy the constancy of the given value of $\zeta$. One should not be surprised by the manipulating of $\tau_s$ for each $\Delta_0$ in order to carry $\zeta = const$ at whole considered surface. In some sense during the study the Lifshitz transition the experimentalists do the same, for example, investigation of the anomalous behavior of thermopower in $Li_{1-x}Mg_x$ alloy in a dependence of $x$ close to transition at $x = 0.19$ (see Ref. [26]).

Therefore, one can conclude that while the Lifshitz transition is governed by the parameter $z = \mu - \mu_c$ ($\mu$ is the chemical potential) [9,13,14,20,21], the driving parameter of the transition under consideration is $\zeta - 1$.

# 3 Smearing of the transition due to spacial fluctuations of the magnetic impurities concentration

In the case of the order parameter varying in space, Eq. (1) for the free energy can be generalized by adding heuristically the corresponding gradient term, like in the Ginzburg-Landau theory, that is

$$
\begin{aligned}
F &= -\frac{N(0)}{2}\left\{a\Delta^2 + \frac{1}{4m}(\nabla\Delta)^2\right\} \\
&= -\frac{N(0)}{2}\left\{a\Delta^2 + \frac{1}{4m}\left(\frac{d\Delta}{d\zeta}\right)^2\left(\frac{d\zeta}{dn_s}\right)^2(\nabla n_s)^2\right\},
\end{aligned}
\tag{8}
$$

where $a = 1 - \frac{\pi}{2}\zeta + \frac{2}{3}\zeta^2$ and we recall that $n_s$ is the concentration of magnetic impurities.

Here we already attributed the variation of the value of order parameter to the spacial inhomogeneity of the paramagnetic impurities distribution, elucidating the corresponding gradient in the last term of Eq. (8). One can neglect the impurities concentration fluctuations until the contribution of the "kinetic energy" remains small in comparison to the superconducting condensation energy.

Correspondingly, comparing the second term in Eq. (8) with the first one and using the expression determing the order parameter $\Delta$ at $T = 0$ given by Eq. (7) we find[1] that the fluctuations of the impurities concentration remain insignificant until

$$
\frac{\nabla n_s}{n_s} \ll \frac{1}{\xi}\left(1 - \frac{\pi}{2}\zeta + \frac{2}{3}\zeta^2\right)\left(\frac{4}{\pi\zeta} - 1\right).
\tag{9}
$$

Here $\xi$ is the superconducting coherence length. This evaluation is valid close to the transition point $\zeta = 1$ (indeed, the limit of superconductor without paramagnetic impurities $\zeta \to 0$ does not make sense in such consideration), i.e.

$$
\frac{d\left[\ln n_s(r)\right]}{dr} \ll \frac{1}{\xi}.
\tag{10}
$$

# 4 Thermoelectric effect

It is well known that the Lifshitz transition in normal metals is accompanied by a giant asymmetric peak in the Seebeck coefficient [26–28]. Despite the opinion prevailing in the early period of the study of superconductivity concerning the vanishing of all conventional thermoelectric properties, today we know that a wide variety of interesting thermoelectric effects can exist in superconductors [29]. Among them is the quantization of the magnetic flux passing through the loop consisting of two different superconductors whose junctions are at different temperatures. As demonstrated in [16] the correction to the integer number of flux quanta appears to depend on the temperature difference and thermoelectric coefficients of the superconductors in their normal state. Hence, one could expect the giant growth of this effect when one of the ring legs is close to the gap-gapless transition.

In order to demonstrate this we will calculate the corresponding quasiparticle contribution to the thermoelectric coefficient following the scheme proposed by Ambegaokar and Griffin to calculate the corresponding thermal conductivity (see Ref. [3]). We perform our calculation in

---

[1] See the Supplemental Material for the detailed derivation of Eqs. (9), (10) and (14) and the low-temperature behavior of the order parameter.

the assumption of validity of the weak enough scattering and applicability of the Born approximation. The thermoelectric coefficient $\alpha$ relating the quasiparticle current to the temperature gradient, can be expressed in the form

$$\alpha = -\frac{eN(0)v_F^2}{4T^2} \int\limits_{-\infty}^{+\infty} \frac{\omega d\omega}{\cosh^2\left(\frac{\beta\omega}{2}\right)} \frac{h(\omega,\Delta,\zeta)}{\text{Im}\left\{\Omega(\omega,\Delta,\zeta) + \frac{i}{2\tau_{tr}} + \frac{i}{\tau_s}\left[1 - h(\omega,\Delta,\zeta)\right]\right\}}, \tag{11}$$

where $\tau_{tr}$ is the transport collision time [1] that enters in the conductivity of a normal metal, $v_F$ is the Fermi velocity and $\beta = \frac{1}{k_B T}$ is the inverse temperature. The functions $h(\omega,\Delta,\zeta)$ and $\Omega(\omega,\Delta,\zeta)$ are given by

$$\frac{\Omega(\omega,\Delta,\zeta)}{\Delta} = \sqrt{u^2-1} - i\zeta, \tag{12}$$

$$h(\omega,\Delta,\zeta) = \frac{1}{2}\left[1 + \frac{|u|^2-1}{|u^2-1|}\right], \tag{13}$$

where we recall that parameter $u$ is defined by Eq. (6).

At temperatures close to zero (large values of $\beta$) the main contribution to the integral in Eq. (11) comes from the low frequencies domain $\omega \lesssim \beta^{-1}$. The numerical analysis of the Eq. (11) shows the dramatic enhancement of the thermoelectric coefficient approaching the transition from the gap side (see Fig. 3). The absolute value of the peak linearly decreases with the temperature, still remaining $\epsilon_F/\Delta$ times larger than the background value.

In the immediate vicinity of the transition one can expand the parameter $u$ and consequently the functions $\Omega(\omega,|\Delta|,\zeta)$ and $h(\omega,|\Delta|,\zeta)$ for small values of $\omega$. In result one can obtain the asymptotic behavior of the thermoelectric coefficient close to the phase transition for both the gap and the gapless states.

From the gap side ($\zeta \to 1-$) one finds[2] that the thermoelectric coefficient takes the form

$$\alpha = \frac{4\sqrt{2}\pi^2}{3e} \frac{T}{\Delta} \frac{\sigma_n \tau_s}{\tau_s + 2\tau_{tr}} \sqrt{1 - \zeta^{\frac{1}{3}}}, \tag{14}$$

where $\sigma_n = \frac{2}{3}N(0)v_F^2 e^2 \tau_{tr}$. Eq. (14) determines the magnitude of the Seebeck coefficient in the gap state. Recalling that the value of Seebeck coefficient in the normal metal is $S_n = \frac{\pi^2 k_B}{3e} \frac{T}{E_F}$ one can find that $S_g$ is giant with respect to the latter by the parameter $E_F/\Delta$:

$$S_g = \frac{\alpha}{\sigma_n} = \frac{2^{5/2}\tau_s}{\tau_s + 2\tau_{tr}} \sqrt{1 - \zeta^{\frac{1}{3}}} \left(\frac{E_F}{\Delta}\right) S_n. \tag{15}$$

When performing the same procedure from the gapless side of the transition it can be disappointing to find $\alpha \equiv 0$. Formally this is related to the oddness of the integrand function over $\omega^2$ in this region. Yet, the obtained result does not mean that the thermoelectric coefficient here turns identically zero: in our expansions we did not retain terms of the order $\omega/E_F$, hence the thermoelectric effect from the right of transition point can be comparable to its normal background.

The results of the numerical calculations of the thermoelectric coefficient based on Eq. (11) are shown in Fig. 3. For the evaluation of $\alpha$ we used the dependence of the order parameter modulus as a function of $\zeta$ at zero temperature given by Eq. (7). For large values of $\beta$ (i.e. in the vicinity of $T = 0$), we assumed the temperature variation of $\Delta$ to be very weak and approximated by Eq. (21)[2].

---

[2]See the Supplemental Material for the detailed derivation of Eqs. (9), (10) and (14) and the low-temperature behavior of the order parameter.

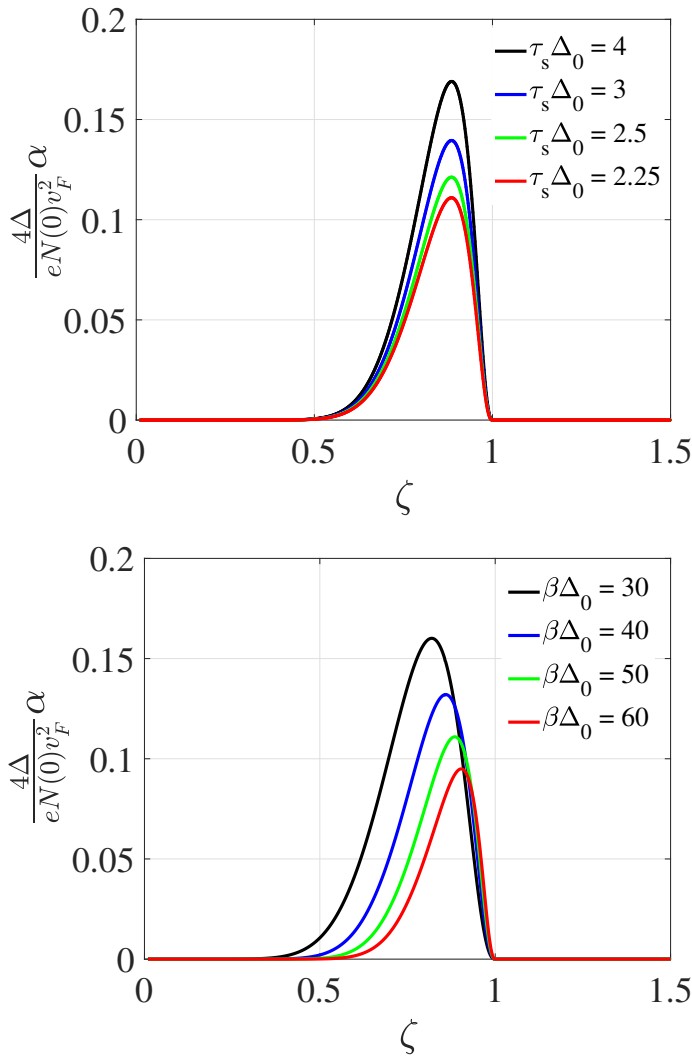

Figure 3: Thermoelectric coefficient as a function of $\zeta$ for different values of $\tau_s\Delta_0$ for the given dimensionless inverse temperature $\beta\Delta_0 = 50$ (up) and for different $\beta\Delta_0$ for the fixed $\tau_s\Delta_0 = 2.25$ (bottom). For both plots $\tau_{tr}\Delta_0 = 1$.

There are several remarkable hallmarks of the found effect. First of all, the thermoelectric coefficient has a giant magnitude in the gap region. Second, the peak is asymmetric, and, third, the peak is shifted from the transition point into the gap domain ($\zeta < 1$) when the temperature increases ($\beta$ decreases). All these features are characteristic also for the Seebeck signal behaviour close to the $2\frac{1}{2}$ phase transitions [8] and can be considered as the smoking gun for the experimental verification of the proposed phenomenon.

We should note that a similar strong enhancement of the thermoelectric coefficient in the presence of magnetic impurities was theoretically predicted in Ref. [30, 31]. However, the authors of Refs. [30, 31] did not relate the revealed giant thermoeffect to the manifestation of the $2\frac{1}{2}$ phase transition. They specified that this phenomenon is caused by violation of the symmetry between electron-like and hole-like excitations due to formation of the subgap Andreev bound states in the vicinity of magnetic impurities [31]. Although different assumptions have been used in Refs. [30] and [31] for the calculation of the thermoelectric coefficient the effect of its enhancement remain the same on the qualitative level.

From the experimental point of view the detection of such a phase transition can be per-

formed by means of placing in magnetic field a ring, one half of which is a gap superconductor with the concentration of magnetic impurities close to the transition value and the other half is an arbitrary superconductor. In this case, provided superconducting contacts are kept at different but low temperatures, anomaly strong thermoelectric current is induced inside the ring and the measured magnetic flux should considerably deviate from the integer values of the magnetic flux quantum $\Phi_0$ [17]. An alternative method for detecting the discussed transition can be a jump in the derivative of the specific heat capacity with respect to the impurity concentration.

The effect of impurity scattering of the DOS dependence on energy and the anisotropy degree has been investigated in Ref. [32] for the case of superconductors with the anisotropic gap. It was shown that presence of a relatively small concentration of impurities leads to the isotropisation of the DOS and decrease of its smearing over energy. With the further increase of the concentration of impurities the region of smearing shrinks to zero. This means that the superconductor becomes effectively isotropic. A full analysis of this problem is outside of the scope of the present paper and therefore left for future studies.

# 5  Conclusions

We have demonstrated that the known for a long time transition between the gap and the gapless states of a superconducting alloy with paramagnetic impurities is the phase transition of the $2\frac{1}{2}$ order. We have shown that the mean-field approximation used in the Abrikosov-Gor'kov theory [1] is very stable: fluctuations of the impurities concentration remain irrelevant in the logarithmic scale. Finally, such a phase transition can be detected by the giant (by the parameter $E_F/\Delta$) thermoelectric effect possessing the characteristic features which would clearly distinguish it from others. We have proposed experiments for detection of such an effect and the subsequent confirmation of the $2\frac{1}{2}$ phase transition nature.

Our theoretical results may help to take a fresh look at recent experiments with light-wave-driven gapless superconductivity [6], for the new interpretation of the theoretically predicted disorder induced transition $s_\pm$-$s_{++}$ states via gapless phase in multi-band superconductors [33, 34] and can be useful for the understanding of gapless color superconductivity in quantum chromodynamics and the string theory [35]. In the case of a dirty multi-band superconductor with increasing of the nonmagnetic impurities concentration, one of the gaps is seen to close, leading to a finite residual DOS, followed by a reopening of the gap. Such a behavior allows to speculate about the Lifshitz origin of $s_\pm$-$s_{++}$ transition. For a color superconductor it was shown that, at zero temperature and small values of the strange quark mass, the ground state of neutral quark matter corresponds to the so-called color-flavor-locked phase. At some critical value of the strange quark mass, there is a transition to the gapless color-flavor-locked phase, where the energy gap in the quasiparticle spectrum is not mandatory [35, 36]. As in the case of multi-band superconductivity one can again speculate about the emergence of the Lifshitz nature of the transition in the phase diagram of the neutral quark matter.

# Acknowledgements

A.V. is grateful to Yu. Galperin, A. I. Buzdin, Aviad Frydman, S. Frolov, S. Bergeret, and O. Dobrovolskiy for valuable and fruitful discussions. C.P. and Y.Y. acknowledge support by the CarESS project. A.V. acknowledges the financial support under the STSM COST Action CA16218 Nanoscale Coherent Hybrid Devices For Superconducting Quantum Technologies.

# 6  Supplemental material for "The Lifshitz nature of the transition between the gap and gapless states of a superconductor"

## 6.1  Spacial fluctuations of the magnetic impurities concentration

Let us perform its evaluation, for simplicity, from the "gap side" of the phase transition. The first derivative in the "kinetic energy" term of Eq. (9) can be easily obtained by direct differentiation of Eq. (10) in the main paper:

$$\left(\frac{d\Delta}{d\zeta}\right) = -\frac{\pi}{4}\Delta. \tag{16}$$

What concerns the derivative $d\zeta/dn_s$ its calculation is more delicate since $\zeta = (\tau_s \Delta)^{-1}$. The scattering lifetime $\tau_s$ is determined by the integral over the solid angle $\Omega$

$$\frac{1}{\tau_s} = \left[ N(0)\frac{S(S+1)}{(2S+1)^2} \int |f_+ - f_-|^2 \, d\Omega \right] n_s = An_s, \tag{17}$$

where $n_s$ is the concentration of the magnetic impurities and $f_+$ and $f_-$ are the scattering amplitudes of an electron with a total angular momentum $S + 1/2$ and $S - 1/2$.

Based on Eq. (17) and the fact that the order parameter $\Delta$ depends on $n_s$ we obtain

$$\frac{d\zeta}{dn_s} = \frac{A}{\Delta} - \frac{A}{\Delta^2}\frac{d\Delta}{dn_s} = \frac{A}{\Delta} - \frac{An_s}{\Delta^2}\frac{d\Delta}{d\zeta}\frac{d\zeta}{dn_s}. \tag{18}$$

Taking into account Eq. (16) one finds

$$\frac{d\zeta}{dn_s} = \frac{1}{n_s}\frac{\zeta}{1 - \frac{\pi\zeta}{4}}. \tag{19}$$

Relating the Cooper pair mass to the coherence length as $\xi^2 = 1/(4ma)$ and returning to Eq. (9) in the main paper one finds that the kinetic energy term in the gap domain ($\zeta < 1$) is expressed as

$$\frac{(\nabla\Delta)^2}{4m} = \frac{\pi^2\xi^2\Delta^2}{16}\frac{\zeta^2}{(1 - \frac{\pi\zeta}{4})^2}\left(\frac{\nabla n_s}{n_s}\right)^2. \tag{20}$$

## 6.2  Low temperature behavior of the order parameter

When the temperature is slightly above $T = 0$ the temperature dependence of order parameter $\Delta$ is given by expressions

$$\Delta(T) = \begin{cases} \Delta(0) - \sqrt{\dfrac{2\pi}{3}}\dfrac{\Delta_g^{1/6}}{(\zeta\Delta(0))^{2/3}}T^{3/2}\left(1 - \dfrac{\pi}{4}\zeta\right)^{-1}e^{-\beta\Delta_g}, \ \zeta < 1 \\[3mm] \Delta(0) - \dfrac{2^{2/3}}{\sqrt{3}}\dfrac{T^{5/3}}{(\Delta(0))^{2/3}}\left(1 - \dfrac{\pi}{4}\right)^{-1}\Gamma\left(\dfrac{2}{3}\right)Z\left(\dfrac{5}{3}\right)\left(1 - 2^{-2/3}\right), \ \zeta = 1 \\[3mm] \Delta(0) - \dfrac{\pi^2}{6}(1 - \zeta^{-2})^{1/2}\left(1 - \dfrac{1}{2}\sqrt{1 - \zeta^{-2}} - \dfrac{\zeta}{2}\arcsin\zeta^{-1}\right)^{-1}\dfrac{T^2}{\Delta(0)}, \ \zeta > 1, \end{cases} \tag{21}$$

where we recall $\Delta_g$ is the energy gap, $\Delta(0)$ is the value of the parameter at $T = 0$, $\Gamma(x)$ is the gamma function and $Z(x)$ is the Riemann zeta function defined by means the capital letter Z to avoid a confusion with the driving parameter $\zeta$.

## 6.3 Asymptotic expressions for the thermoelectric coefficient

In the vicinity of the zero temperature or for the large values of $\beta$ the contribution to the Eq. (13) in the main paper gives the low order frequencies. Such a restriction allows to obtain several useful asymptotics for the thermoelectric coefficients from the gap and the gapless side of the phase transition. The starting point is the approximated expression for the parameter $u$ in the case of the small $\omega$.

### 6.3.1 Gap state

For the gap regime, where $\zeta < 1$ we have

$$\frac{\omega - \Delta_g}{\Delta} = -\frac{3}{2}\zeta^{-\frac{2}{3}}\left(1 - \zeta^{\frac{2}{3}}\right)^{\frac{1}{2}}(u - u_0)^2, \tag{22}$$

where

$$\frac{\Delta_g}{\Delta} = \left(1 - \zeta^{\frac{2}{3}}\right)^{\frac{3}{2}}, \tag{23}$$

and

$$u_0 = \left(1 - \zeta^{\frac{2}{3}}\right)^{\frac{1}{2}}. \tag{24}$$

Substitution of Eqs. (23)-(24) into Eq. (22) yields an equation for $u$ with the solution for $\Delta_g = 0$, i.e. when $\zeta \to 1$

$$u = \sqrt{1 - \zeta^{\frac{2}{3}}} \pm \frac{1}{3}\frac{\sqrt{6\zeta^{\frac{2}{3}}\left[\left(1 - \zeta^{\frac{2}{3}}\right)^{\frac{3}{2}} - \frac{\omega}{\Delta}\right]}}{\left(1 - \zeta^{\frac{2}{3}}\right)^{\frac{1}{4}}} \approx \sqrt{2\left(1 - \zeta^{\frac{1}{3}}\right)} \pm \frac{1}{3}i\frac{\sqrt{6\left[\frac{\omega}{\Delta} - 2\sqrt{2}\left(1 - \zeta^{\frac{1}{3}}\right)^{\frac{3}{2}}\right]}}{2^{\frac{1}{4}}\left(1 - \zeta^{\frac{1}{3}}\right)^{\frac{1}{4}}}. \tag{25}$$

Based on Eqs. (25) for the parameter $u$ one can write the expression for functions $\Omega(\omega, \Delta, \zeta)$ and $h(\omega, \Delta, \zeta)$ that are entered in Eq. (13) in the main paper for the thermoelectric coefficient in the main text. Introducing a new parameter $z = \sqrt{1 - \zeta^{\frac{1}{3}}}$ near the the phase transition we have

$$\frac{\Omega(\omega, \Delta, \zeta)}{\Delta} = \sqrt{2z^2 + \frac{\sqrt{2}w}{3z} - 1 + i\frac{2}{3}2^{\frac{1}{4}}\sqrt{6zw}} - i, \tag{26}$$

and

$$h(\omega, \Delta, \zeta) = \frac{1}{2}\left[1 + \frac{2z^2 + \frac{\sqrt{2}w}{3z} - 1}{\sqrt{\left(2z^2 + \frac{\sqrt{2}w}{3z} - 1\right)^2 + \frac{8\sqrt{2}}{3}zw}}\right], \tag{27}$$

where $w = \frac{\omega}{\Delta} - 2\sqrt{2}z^3$ and for the extraction of the square root of a complex number in Eq. (26) the well-known formula is applied

$$\sqrt{a + ib} = \pm\sqrt{\frac{\sqrt{a^2 + b^2} + a}{2}} \pm i\,\mathrm{sgn}\,b\sqrt{\frac{\sqrt{a^2 + b^2} - a}{2}}. \tag{28}$$

Using Eq. (26) and (27) one can expand in series for small $\omega$ the part of the integrand in Eq. (13) in the main paper

$$\frac{h(\omega, \Delta, \zeta)}{\mathrm{Im}\left\{\Omega(\omega, \Delta, \zeta) + \frac{i}{2\tau_{tr}} + \frac{i}{\tau_s}(1 - h(\omega, \Delta, \zeta))\right\}} \approx \Upsilon_0(\zeta, \tau_{tr}, \tau_s) + \Upsilon_1(\zeta, \tau_{tr}, \tau_s)\frac{\omega}{\Delta}, \tag{29}$$

where $\Upsilon_0(\zeta, \tau_{tr}, \tau_s)$ and $\Upsilon_1(\zeta, \tau_{tr}, \tau_s)$ are some function that we do not present explicitly due to their very cumbersome expressions. However, one can also perform the expansion in series of this function for $z = 0$ or $(\zeta = 1)$ to simplify further analytical calculations

$$\Upsilon_1(\zeta, \tau_{tr}, \tau_s) \approx \frac{4\sqrt{2}}{3} \frac{\tau_s \tau_{tr}}{\tau_s + 2\tau_{tr}} z. \tag{30}$$

Therefore, combining Eqs. (29) and (30) finally we obtain asymptotic expression for the thermoelectric coefficient close to the phase transition from the gap side

$$
\begin{aligned}
\alpha &= \frac{8\sqrt{2}}{3} \frac{eN(0)Tv_F^2}{\Delta} \frac{\tau_s \tau_{tr}}{\tau_s + 2\tau_{tr}} z \int_{-\infty}^{+\infty} \frac{\omega^2 d\omega}{\cosh^2\left(\frac{\omega}{2T}\right)} \\
&= \frac{4\sqrt{2}\pi^2}{9} \frac{eN(0)Tv_F^2}{\Delta} \frac{\tau_s \tau_{tr}}{\tau_s + 2\tau_{tr}} z \\
&= \frac{4\sqrt{2}\pi^2}{9} \frac{eN(0)Tv_F^2}{\Delta} \frac{\tau_s \tau_{tr}}{\tau_s + 2\tau_{tr}} \sqrt{1 - \zeta^{\frac{1}{3}}}.
\end{aligned}
\tag{31}
$$

### 6.3.2 Gapless state

In the case of the gapless regime, where $\zeta > 1$ the expansion of $u$ is given by

$$u = i\sqrt{\zeta^2 - 1} + \zeta^2(\zeta^2 - 1)^{-1}\frac{\omega}{\Delta} + \dots. \tag{32}$$

This allows to obtain in the same way the expression for functions $\Omega(\omega, \Delta, \zeta)$ and $h(\omega, \Delta, \zeta)$ that are entered in Eq. (13) in the main paper for the thermoelectric coefficient in the main text. As in the previous case introducing a new parameter $z = \sqrt{\zeta - 1}$ near the the phase transition one can write

$$\frac{\Omega(\omega, \Delta, \zeta)}{\Delta} = \sqrt{2z + \frac{1}{4}\frac{1}{z^2}\frac{\omega^2}{\Delta^2} - 1 + i\frac{\sqrt{2}}{\sqrt{z}}\frac{\omega}{\Delta}} - i, \tag{33}$$

and

$$h(\omega, \Delta, \zeta) = \frac{1}{2}\left[ 1 + \frac{2z + \frac{1}{4z^2}\frac{\omega^2}{\Delta^2} - 1}{\sqrt{\sqrt{\left(2z + \frac{1}{4}\frac{1}{z^2}\frac{\omega^2}{\Delta^2} - 1\right)^2 + \frac{2}{z}\frac{\omega^2}{\Delta^2}}}} \right]. \tag{34}$$

Based on Eq. (33) and (34) we expand in series for small $\omega$ the part of the integrand in Eq. (13) in the main paper

$$\frac{h(\omega, \Delta, \zeta)}{\text{Im}\left\{ \Omega(\omega, \Delta, \zeta) + \frac{i}{2\tau_{tr}} + \frac{i}{\tau_s}(1 - h(\omega, |\psi|, \zeta)) \right\}} \approx \Theta_0(\zeta, \tau_{tr}, \tau_s) + \Theta_1(\zeta, \tau_{tr}, \tau_s)\frac{\omega^2}{\Delta^2}. \tag{35}$$

Due to long expressions for functions $\Theta_0(\zeta, \tau_{tr}, \tau_s)$ and $\Theta_1(\zeta, \tau_{tr}, \tau_s)$ we do not provide them in the explicit form. Nevertheless, since the expansion in series given by Eq. (35) contains only the even degree of $\omega$ it is easy to understand that the integrand in Eq. (11) in the main paper is the odd function of $\omega$ and, hence, the integral is equal to zero.

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
