# Peer review of "Topological phase transition between the gap and the gapless superconductors"

_SciPost Physics, doi:SciPost Phys. Core 5, 009 (2022)_

## Round 1 · Referee Report · Anonymous (Referee 1) · 2021-10-12

Strengths
Weaknesses
Report
Consequently, I do not recommend the manuscript for publication in SciPost Physics. Below I expand on the points mentioned above and provide questions and suggestions to be addressed in the manuscript.
Requested changes
(1) The Authors offer a reinterpretation of this transition in terms of a topological invariant - the Euler characteristic calculated for the DOS surface. Are any physical quantities of the system uniquely determined by this invariant?
$\bullet$ Is there any interplay of the Euler characteristic with the known cases of topology in superconductors (e.g. p+ip state in 2D)? Being a property of DOS, I believe that it will not distinguish between trivial (s-wave) and topological (p+ip in 2D) states.
(2) All calculations were performed for the case of an ideally isotropic gap. How will gap anisotropy affect the results? In particular, for strong anisotropy with deep minima, can the behavior of DOS and free energy near the transition change qualitatively?
(3) The Authors refer to Abrikosov-Gor'kov theory for the disordered superconductivity throughout the text - does that imply that Born approximation for scattering is used? In particular, does the justification for the stability of the mean-field description given in "Smearing of the transition due to spacial fluctuations of the magnetic impurities concentration", rely on Born approximation? How will rare region effects and the presence of impurity bound states affect the argument?
(4) The relation to previous works on thermoelectric coefficients in superconductors with magnetic impurities has to be discussed in more detail. In Ref. 33, multiple scattering effects were considered beyond the Born approximation, so it appears that the results of Ref. 33 are more general. In Ref. 32, on the other hand, results in Born approximation were reported and Eq. (11) and (12) there are indeed quite similar to Eq. (13) and (8) of the current work. However, the denominators in Eq. (13) and Eq. (11) of Ref. 32 appear different - could the Authors explain this difference?
$\bullet$ Since results at finite $T$ are reported in Fig. 3, was $\Delta(\zeta)$ calculated self-consistently for finite $T$ or were the zero-temperature expressions used? If the latter is true, this has to be mentioned and justified.
Minor comments/suggestions:
--- Fig. 1 misses a color scale; the drastic change from purple to blue seems to suggest a jump, which can be confusing to readers, since the transition is actually continuous.
--- Mentions of the applications to $s_{++}/s_{\pm}$ transition and color superconductivity in QCD and string theory are not really substantiated or discussed. The Authors should either provide a discussion of what new physics can their approach reveal in those systems or refrain from stating the connection (at least in the abstract and conclusion).
--- Some links in citations are not working (e.g. 17,18); Ref. 17 links to the same URL as Ref. 18; Ref. 18 is missing the journal information.
Please find our response in the file attached

Anonymous on 2021-09-21 [id 1771]
A question which the authors may want to address, to help me out of my confusion. Figures 2c and 2d show a topological change in the Fermi surface before and after the gap closing transition. It is argued that an equivalent topological transition appears in the density of states as a function of frequency , where figure 2c corresponds to figure 2a and figure 2d to 2b. I am unsure how this comparison works.
Figure 2c is at a fixed value of the parameter $\zeta<1$, say $\zeta=1/2$, while figure 2d is for a fixed $\zeta>1$, say $\zeta=3/2$ and these two figures are indeed topologically distinct. Now to compare with figures 2a and 2b I would look at the curves $N(\omega)$ versus $\omega$ at $\zeta=1/2$ and $\zeta=3/2$. These two curves are topologically identical, both show a cusp at $\omega=0$. So in what sense does the topological distinction of figures 2c,2d carry over to figures 2a,2b?
I ask because this topological correspondence is the central point of the paper. Basically I don't see how the 2D surfaces of figures 2c,2d correspond to 1D cuts in figures 2a,2b.
I notice that in an earlier version of this manuscript on arXiv:2105.01934v1 there was indeed a comparison of surfaces at fixed $\zeta$. In this version figures 2a,b contain $\zeta$ on one of the axes, so the comparison is to a surface cut and then I don't see a topological distinction.
Author: Yuriy Yerin on 2021-09-23 [id 1779]
(in reply to Anonymous Comment on 2021-09-21 [id 1771])Thank you so much for the valuable comment. You have raised a crucial question. We provided a detailed explanation in the file attached.
Attachment:
reply_comment1.pdf
Anonymous on 2021-10-13 [id 1844]
(in reply to Yuriy Yerin on 2021-09-23 [id 1779])Thank you for the helpful answer. A smaller question. Near figure 2 I read " the Euler characteristic changes from χ = 2 (gap state) to χ = 1 (gapless state)." At the end of the paper, instead, it says "the Euler characteristic changes from χ = 0 (gap state) to χ = 1 (gapless state)."
This is contradictory, and moreover I think both statements are incorrect.
If I look at figures 2c and 2d, and take into account that I can translate the Fermi surface by a reciprocal lattice vector, I see a surface that is topologically equivalent to a sphere in figure 2c and to a torus in figure 2d, so the Euler characteristic would change from χ = 2 (gapped state) to χ = 0 (gapless state).
Author: Yuriy Yerin on 2021-10-13 [id 1846]
(in reply to Anonymous Comment on 2021-10-13 [id 1844])Many thanks for the careful reading of the manuscript.
The statement at the end of the manuscript is a footprint of the first revision https://arxiv.org/pdf/2105.01934v1.pdf
We agree that is incorrect. Thank you again.
However, we partially agree that the Euler characteristic would change from χ = 2 (gapped state) to χ = 0 (gapless state). According to our numerical calculations, the Euler characteristic, which unambiguously changes during the transition from the gapped state to the gapless state, in the latter case has exactly 1 and not 0 due to the singularity in the form of the cuspidal edge in Figure 2b.

---

## Round 1 · Referee Report · Carlo Beenakker (Referee 2) · 2021-10-14

Report
The claim is not substantiated in any way, it is merely stated that χ=2 for ζ<1 and χ=1 for ζ>1. Since it is a central claim, there should be no ambiguity. For starters, I am not even sure that the Euler characteristic is well defined on the surface N(ω,ζ), because the cuspidal edge at ζ=1 has a divergent Gaussian curvature. The poles at ζ=0 are also worrisome. If these poles can be regularised, then I would think that both the surfaces at ζ<1 and ζ>1 can be contracted to a point, which would imply that χ=1 for both surfaces and there is no topological transition.
Please find our response in the file attached
Attachment:

---

## Round 2 · Referee Report · Anonymous (Referee 1) · 2021-12-6

Report

In the new version of the manuscript, the Authors have added a number of clarifications regarding the scope of their calculations as well as the broader context of the study. However, I believe that the most important comment from my previous report ("Are any physical quantities of the system uniquely determined by this invariant?") has not been addressed properly. Let me elaborate on it to avoid misunderstanding.

While the Chern number of quantum Hall states or topological superconductors can be related to the Euler characteristic, the crucial property of these states is the existence of a physical observable (Hall or thermal Hall conductivity) that is determined by the topology only (and the universal constants), i.e. "quantized". A recent example, where Euler characteristic has been related to physical observables can be found in arXiv:2108.05870. The authors suggest ``the number of gaps in a system under consideration" as such a quantity - however, no way of actually measuring it is suggested. One can think of measuring DOS at $\omega=0$ via STM; however, it is known that in actual experiments (see e.g., Phys. Rev. 137, A557 (1965); Phys. Rev. Lett. 53, 2437 (1984)) a non-zero density of states at $\omega=0$ appears routinely even where it is not expected from an Abrikosov-Gor'kov theory standpoint. While the origin of this behavior is not fully clear (see e.g., Phys. Rev. Lett. 105, 026803 (2010), Phys. Rev. B 94, 144508 (2016)), it effectively rules out STM. Spectroscopic measurements, such as optical conductivity, would be contaminated at finite temperatures by quasiparticles. Taking into account in addition that the gapless superconductivity is expected to occur in an extremely narrow parameter regime, it does not appear that the topological characteristic can manifest itself robustly in experiments, severely limiting the impact of the present work.

The calculation of the thermoelectric coefficient, the second major result of the work, as the Authors acknowledge in their Reply, has been considered previously for a more general case: "the more general consideration in Ref. 34". Taking the points above into account, I can not recommend the current manuscript for publication in SciPost Physics, given that the criterion for this journal is to ``provide details on groundbreaking results". However, as the paper does explore interesting theoretical connections and provide guidelines for developments in thermoelectric effect, I believe that this paper is appropriate for publication in SciPost Core.

Additionally, there is a technical comment I would like to make. I believe that the statement that "at $\omega=0$, $\zeta=1$ the Gaussian curvature is not diverged and equal to zero" is not, strictly speaking, correct, when Eq. (17) for DOS is used. While $K=0$ when the limit $\zeta\to0$ is taken before $\omega \to0$, this is not so if the order of limits is reversed. Indeed for $\zeta\to 1,\omega=0$ one gets $\frac{\partial N}{\partial \omega},\frac{\partial^2 N}{\partial \omega \partial\zeta} = 0$, $\frac{\partial N}{\partial \zeta} \propto (\zeta-1)^{-1/2}$, $\frac{\partial^2 N}{\partial \zeta^2} \propto (\zeta-1)^{-3/2}$, $\frac{\partial^2 N}{\partial \omega^2} \propto (\zeta-1)^{-5/2}$. Using these expressions in (18) one gets $K\propto (\zeta-1)^{-2}$ that does diverge as $\zeta\to1$. While it is likely that the integral over both $\zeta$ and $\omega$ still converges near $\omega=0$, $\zeta=1$, I believe that the details of the analysis leading to this conclusions should be presented (and possible issues with the order of limits also considered for the $\omega=\Delta_g$, $\zeta\to 0$ case).
  • validity: good
  • significance: ok
  • originality: good
  • clarity: high
  • formatting: good
  • grammar: good

Author:  Yuriy Yerin  on 2021-12-10  [id 2022]

(in reply to Report 1 on 2021-12-06)
Category:
reply to objection

Please find our response in the file attached

Attachment:

Referee1_new_reply_final.pdf

---

## Round 2 · Author Response

Dear Editor.
Thank you for your e-mail message of October 14, 2021 and sending us the referee reports regarding the manuscript “Topological phase transition between the gap and gapless superconductors” by Yuriy Yerin, Caterina Petrillo, and A. A. Varlamov.

Please find enclosed the revised version of our manuscript, the answers to the Referee’s questions and comments, and the list of changes performed. For the sake of convenience to follow all changes have been made we also attached the pdf file where all improvements corresponding to the Referee questions and comments are highlighted by blue colour.
In view of recognition by the Referee of the hotness of the topic and originality of our findings we believe that the revised version will be approved by him and will be suitable for publication in SciPost.

---

## Round 2 · List of Changes

1. Additional arguments for the existence of the topological transition have been provided.
  2. The supplemental material with details of the Euler characteristic calculation has been extended.
  3. New parts of the manuscript are highlighted in blue.

---

## Round 3 · Author Response

Dear Editor,
Thank you for giving us another opportunity to improve the manuscript. Following the recommendations of one of the referees, we removed from the paper the mentioning of the possibility of characterization of the gap-gapless transition by means of the topological invariant in the form of the Euler characteristic. Correspondingly, we removed the word “topological” also from the title of the paper.
Concerning the place of publication, we would like to draw your attention to the recommendation of one of the referees that the manuscript is worthy of publication in SciPost. This work has already been presented at several conferences and has attracted lively interest not only from theorists, but also from experimentalists.
Finally, the paper is not only about the Lifshitz nature of the transition in the Abrikosov-Gorkov theory, but also concerns the possible application of the found circumstance to other areas of physics: namely, the change of symmetry of the order parameter in the two-gap superconductor and quantum chromodynamics.
In view of these facts, we hope to see our paper published at the pages SciPost.
Best regards and Happy New Year
Yuriy Yerin
Caterina Petrillo
Andrey Varlamov

---

## Round 3 · List of Changes

1. Following the recommendations , we eliminated from the paper the mentioning of the possibility of characterization of the gap-gapless transition by means of of the Euler characteristic.
  2. We removed the word “topological” also from the title of the paper and correspondingly we changed the title of the paper.
  3. We stressed out on the emergence of the cuspidal edge at the density of states surface $N(\omega,\Delta_0)$ ($\Delta_0$ is the value of the superconducting order parameter in the absence of magnetic impurities) and the occurrence of the catastrophe phenomenon at the transition point.

---

## Editorial Decision

published